# Forest Structure Drives Fuel Moisture Response across Alternative Forest States

**Tegan P. Brown** [1,*], **Assaf Inbar** [1], **Thomas J. Duff** [1,2], **Jamie Burton** [3], **Philip J. Noske** [1], **Patrick N. J. Lane** [1] **and Gary J. Sheridan** [1]

1   School of Ecosystem and Forest Sciences, Faculty of Science, The University of Melbourne, Baldwin Spencer Building, Parkville, VIC 3010, Australia; assaf.inbar@unimelb.edu.au (A.I.); thomas.duff@cfa.vic.gov.au (T.J.D.); pnoske@unimelb.edu.au (P.J.N.); patrickl@unimelb.edu.au (P.N.J.L.); sheridan@unimelb.edu.au (G.J.S.)
2   Bushfire Management, Country Fire Authority, Burwood, VIC 3125, Australia
3   School of Ecosystem and Forest Sciences, Faculty of Science, The University of Melbourne, Creswick, VIC 3363, Australia; jamie.burton@unimelb.edu.au
*   Correspondence: tpbrown@student.unimelb.edu.au; Tel.: +61-403-911-133

**Abstract:** Climate warming is expected to increase fire frequency in many productive obligate seeder forests, where repeated high-intensity fire can initiate stand conversion to alternative states with contrasting structure. These vegetation–fire interactions may modify the direct effects of climate warming on the microclimatic conditions that control dead fuel moisture content (FMC), which regulates fire activity in these high-productivity systems. However, despite the well-established role of forest canopies in buffering microclimate, the interaction of FMC, alternative forest states and their role in vegetation–fire feedbacks remain poorly understood. We tested the hypothesis that FMC dynamics across alternative states would vary to an extent meaningful for fire and that FMC differences would be attributable to forest structural variability, with important implications for fire-vegetation feedbacks. FMC was monitored at seven alternative state forested sites that were similar in all aspects except forest type and structure, and two proximate open-weather stations across the Central Highlands in Victoria, Australia. We developed two generalised additive mixed models (GAMMs) using daily independent and autoregressive (i.e., lagged) input data to test the importance of site properties, including lidar-derived forest structure, in predicting FMC from open weather. There were distinct differences in fuel availability (days when FMC < 16%, dry enough to sustain fire) leading to positive and negative fire–vegetation feedbacks across alternative forest states. Both the independent ($r^2$ = 0.551) and autoregressive ($r^2$ = 0.936) models ably predicted FMC from open weather. However, substantial improvement between models when lagged inputs were included demonstrates nonindependence of the automated fuel sticks at the daily level and that understanding the effects of temporal buffering in wet forests is critical to estimating FMC. We observed significant random effects (an analogue for forest structure effects) in both models ($p < 0.001$), which correlated with forest density metrics such as light penetration index (LPI). This study demonstrates the importance of forest structure in estimating FMC and that across alternative forest states, differences in fuel availability drive vegetation–fire feedbacks with important implications for forest flammability.

**Keywords:** fire; climate change; alternative state; feedbacks; obligate seeder; *Eucalyptus regnans*; fuel moisture content; fuel availability

## 1. Introduction

Fire is a critical process in many ecosystems globally that influences the distribution, composition and successional stage of vegetation communities [1]. While fire is important for the maintenance of many ecosystems, altered fire regimes can have negative impacts [2]. Climate warming is elevating fire danger in many locations across the globe [3–5] and increasing temperatures coinciding with more variable rainfall are expected to increase

fire frequency in southeastern Australia [6,7]. Repeated high-intensity fires can alter the successional pathways of forest communities by overwhelming the utility of fire adaptive traits [8], which can lead to abrupt shifts in ecosystem composition and forest structural properties [9–11]. Given the inevitable nature of climate warming, a key challenge in this region is determining whether shifts in forest composition would amplify or dampen effects on fire activity in forests.

Fire-adaptive traits enable species persistence at a landscape level and are broadly characterised along a continuum between resprouting and obligate seeding [12]. While resprouting forests can persist in maturity through fire and are considered fire tolerant [13], obligate seeding forests are generally killed by high-intensity fire and persist through mass regeneration from seed [14]. They are considered fire sensitive because if regenerating juveniles are burnt before reproductive age, local extinction can occur [13] and the dominant vegetation community may transition to one more adapted to short-interval fires [11,15]. The potential for climate-induced changes in fire frequency to drive forest conversion is a global challenge and has been recognised in high-altitude forests in Patagonia [16], tropical forests in the Amazon [17], boreal forests of North America [10,18–20] and Australian eucalypt forests [21–24].

*Eucalyptus regnans* (Mountain Ash) (F.Muell.)) is an obligate seeding dominant over-storey eucalypt species present in southeastern Australia. *E. regnans* forests are highly valued for ecosystem services such as water quality [25], tourism [26], timber [27] and biodiversity values [28]. Changes to fire frequency have the potential to reduce the capacity of *E. regnans* forests to provide these important services [29], yet our understanding of the mechanisms through which climate warming will directly and indirectly affect forests is currently inadequate.

The potential for forest conversion to be sustained is moderated by fire–vegetation feedbacks [30]. While feedbacks exist in nature on a continuum, here we use a discrete definition of positive, negative or no feedback in the context of dead fine fuel moisture content (FMC). FMC is the mass of water per unit mass of dry material [31]. In forests, live and dead vegetation are fuel to a fire, and herein, we define FMC as dead surface fuels, which are a key determinant of fire risk in forests [32]. *E. regnans* are high-productivity forest systems, which almost always have enough fuel to sustain fire [33]. Consequently, fire activity, and therefore the potential for fire–vegetation feedbacks, is more closely associated with the moisture content of fuel than the amount [34]. System changes that result in lower FMC are considered positive feedbacks, while higher FMC is considered a negative feedback (Figure 1). If no change in FMC is observed (in addition to those resulting from climate warming directly), then no feedback is present. Negative feedbacks typically stabilise a system, while positive feedbacks can amplify the effects of climate warming [35] and have the potential to cause and sustain abrupt changes in forest composition and structure [36]. Given the inherent value of *E. regnans* forests, and the high consequence of their conversion to alternative forest states, understanding the potential for, and stability of, such conversions is an important knowledge gap in these forests.

*E. regnans* forests are adapted to low-frequency, high-intensity fire (~75–150+ years) [37] and reach reproductive maturity at approximately 15–20 years. Therefore, they are vulnerable to reproductive failure if burned below this threshold [38,39]. In the southeastern Australian uplands, multiple short-interval fires have created a mosaic of alternative forest states with strongly contrasting structural properties across the landscape previously dominated by *E. regnans* [40]. This includes stands of pure *Acacia dealbata* (Link.) [41] and mixed non-eucalypt forests dominated by understorey species such as *Pomaderris aspera* (Sieber ex DC.) and tree ferns (e.g., *Dicksonia antartica* (Labill.) and *Cyathea australis* (R.BR.)). The successional sequence of forest distribution, disturbance, abatement and recolonisation for wet Eucalypt forest have been described in detail [21,24] and are a natural part of their life cycle. However, under climate warming conditions, with associated increases to fire frequency, there is considerable uncertainty regarding the strength, and therefore

permanency, of potential fire–vegetation feedbacks and alternative forest states. Figure 1 outlines a conceptual model of potential fire–vegetation feedbacks considered in this study.

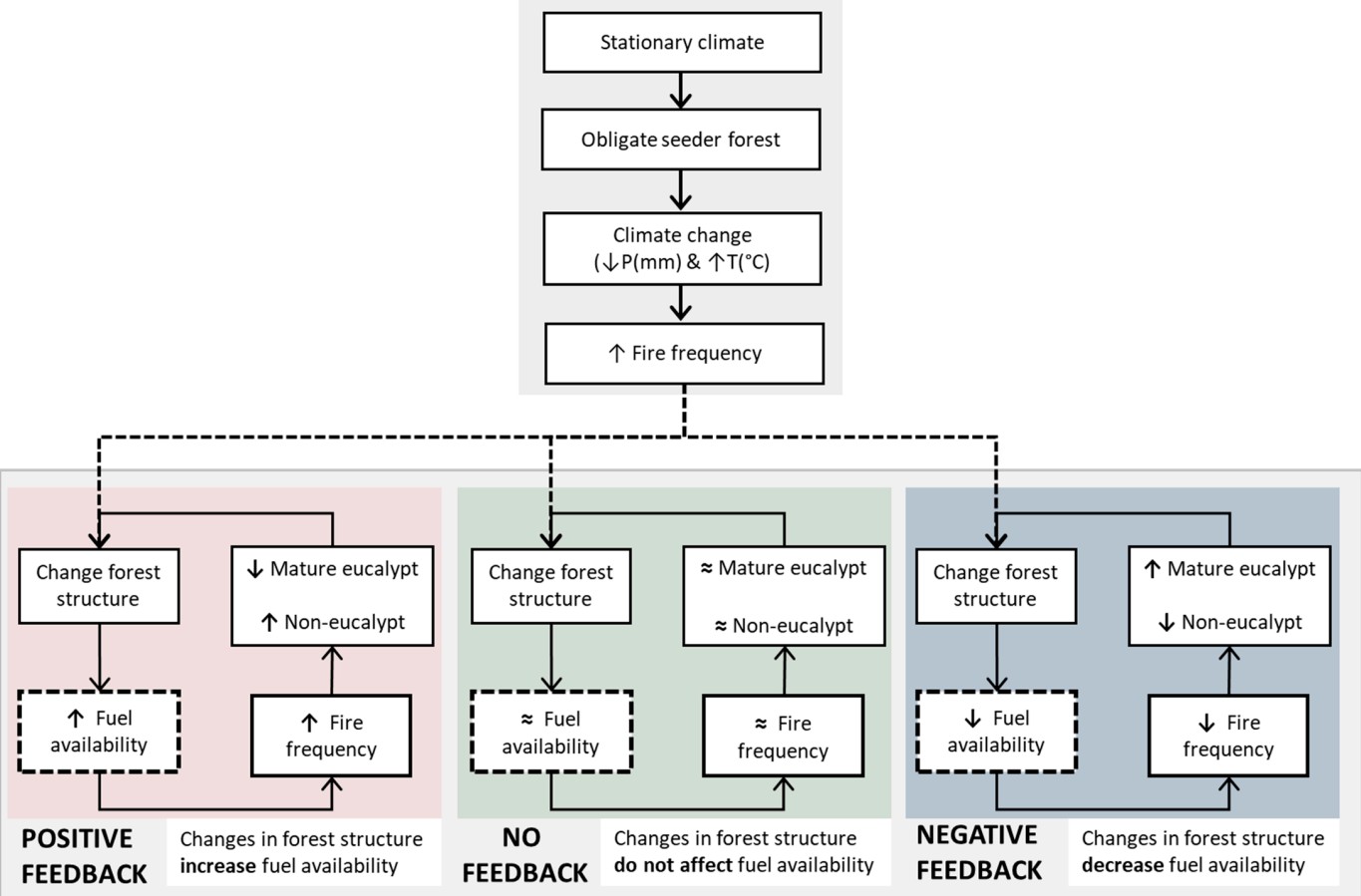

**Figure 1.** A conceptual model of potential feedback processes in response to climate-change-induced increases to fire frequency in obligate seeding wet forests. Three alternative feedback processes are depicted: positive (pink), no feedback (green) and negative (blue). Dashed lines represent the components of this conceptual model under investigation in this study.

Previous research across the alternative forest states depicted reported divergent fuel moisture responses, supporting the presence of fire–vegetation feedbacks [24]. However, the site-level conditions causing this variability were not discernible. The moisture content of dead fine fuels is primarily a function of the weather conditions at the fuel surface, also known as the microclimate, which is strongly influenced by forest structure [42]. Consequently, a considerable knowledge gap remains regarding the key microclimate variables, and the extent to which alternative forest states mediate fuel moisture conditions across these different forest systems.

Forest canopies buffer microclimate extremes, functioning as a thermal insulator for biotic and abiotic ecosystems in the understorey [43–45]. The capacity of forests to buffer microclimate have been reported for temperature [43,46–48], humidity and vapor pressure deficit [42,49–51], while the attenuating effects of forest canopies are well established for radiation [52], wind speed [53–55] and throughfall [56,57]. While the ecological implications of microclimate buffering are widely reported [42,58–60], the application of this paradigm to fuel moisture modelling is currently limited. Existing methodologies for estimating FMC utilise above-canopy products derived at the point- or landscape-scale [61]. While these data sources provide high-level insight, our ability to characterise forest microclimate, and subsequently FMC, at a scale meaningful for fire ignition, and propagation is limited

because understorey conditions cannot be directly measured from above the canopy [47]. However, we currently lack a detailed understanding of the dominant structural properties controlling microclimate buffering in forests and the subsequent effect on FMC. In turn, this limits our capacity to understand the effects of climate warming, and the potential for vegetation-fire–climate feedbacks in these highly valued forest systems.

FMC is a key determinant of fire risk [32,62] and resultant behaviour [63]. It is a function of conditions at the fuel–atmosphere interface (microclimate) and intrinsic fuel properties than influence the time-lag of moisture exchange [64]. Fuels theoretically gain or lose moisture exponentially as FMC approaches equilibrium with the microclimate [65]. In forested settings, 10 h automated fuel sticks are often used as an analogue of FMC to collect continuous timeseries data [66]. The time-lag of these sticks suggests that the moisture exchanged after 10 h correspond to 63% of the difference between initial and equilibrium microclimate conditions, and near-complete moisture exchange is expected within 24 h [67].

While automated fuel sticks are a useful tool for collecting continuous FMC timeseries, they are limited to the point-scale where they are installed. Consequently, models are required to extrapolate across landscapes. FMC models are typically empirically or physically based. Empirical models utilise statistical relationships between weather and observations to derive FMC [68], while physical-based models, which include equilibrium moisture content [69,70] and process models [71,72], describe the fundamental processes governing the movement of water into and out of fuels. FMC models can be account-keeping, where FMC at time $t$ is a function of microclimate and $FMC_{t-1}$ or instantaneous. Despite the substantial knowledge base surrounding FMC models, there is no established paradigm for the site and weather conditions under which an account keeping or instantaneous model is most appropriate. While instantaneous models may be suitable in dry forests, where the effects of canopy are limited [73], dense canopies characteristic of wet forests disconnect understorey fuels from the prevailing weather conditions [43], such that a system change above the canopy is not directly or immediately replicated in below-canopy conditions. In turn, this temporal buffering facilitated by the canopy may generate a lagged response in FMC dynamics. However, there are limited studies modelling subcanopy FMC from open weather that explicitly account for forest structure effects on the weather inputs. Consequently, disentangling these effects remains a key knowledge gap.

Our study aimed to test the hypothesis that, across alternative forest states, dead fuel moisture would be different to an extent meaningful for fire ignition and spread, and that FMC variability, and consequently the strength of fire–vegetation feedbacks, could be attributed to quantifiable differences in forest structural characteristics. Specifically, we addressed the following questions:

1.  Are there differences in fuel moisture content across alternative forest states?
2.  Can FMC at the forest floor (as represented by 10-h fuel moisture sticks) be accurately modelled from open-weather conditions?
3.  Which forest properties have the greatest influence on subcanopy FMC?

## 2. Materials and Methods

### 2.1. Overview

FMC and prevailing weather were monitored at seven forested sites (instrumented) and two adjacent open-weather stations (control). Forest structural metrics were described using airborne light detection and ranging (lidar) data, which was also used to check that the forest structure at each instrumented site was representative of that alternative state forest across the broader landscape. We developed statistical models using a mixed model approach (whereby FMC measurements were nested within sites) to predict daily mean FMC from open-weather-derived predictor variables. Linear regression was used to evaluate relationships between the site level random effect model intercepts and instrumented site properties. These regressions were then used to interpret the importance of each structural property for estimating FMC from open-weather conditions at the site level.

## 2.2. Study Area

The study was conducted in the Central Highlands of Victoria in southeastern Australia, on the traditional lands of the Wurundjeri and Taungurung people [74]. Seven field sites (instrumented) and two open (control) weather stations were established in Wet Forest (Ecological Vegetation Class 30) in the Highlands—Southern Fall Bioregion [75].

The climate in this region is temperate (Köppen–Geiger type Cfb), with warm dry summers and cool wet winters [76]. Mean annual rainfall for the area is 1000–1400 mm $y^{-1}$, and mean daily maximum (January) and minimum (July) temperatures are 25.4 and 11.8 °C, respectively [77]. The soils in this region are deep and fertile [40].

### 2.2.1. Site Selection

Study sites were selected to be as similar as possible except for fire history, to control for other factors contributing to FMC variability. All sites are south facing, have similar elevation and mean annual rainfall (Table 1), and are not located on ridgelines or in soil moisture convergence zones. As outlined in Figure 2, six of the forested sites are located near Powelltown, while the multi-cohort *E. regnans*$_{260}$ site and its open-weather station is located near Maroondah as this forest type, while controlling for other landscape factors was not available near the Powelltown sites. The forest mosaic utilised in this study resulted from a series of stand-replacing fires in 1759, 1926, 1932, 1939, 2009 and 2017 (Table 1) which led to the arrangement of different forest ages and alternative states in the region. Four sites are dominated by *E. regnans* of different age classes, and three are non-eucalypt forest types. The *E. regnans*$_2$ (2 years since stand-replacing disturbance) site was clear-fell harvested and regenerated through burning, which is a common silvicultural technique in the area [78]. *E. regnans*$_{260}$ is located in the Myrtle Creek catchment of Maroondah Reservoir and regenerated from stand-replacing fire in 1759 [79]. The area was subsequently burnt by low-severity fire in 1939 and 2009 that did not result in stand replacement. Consequently, the *E. regnans*$_{260}$ site represents multiple forest cohorts, which is not uncommon for *E. regnans* forests of this age [14,80]. The same 1939 fire in Maroondah burnt forest near Powelltown, which included areas previously impacted by stand-replacing fires in 1926 and 1932. Due to multiple fires in short succession, large areas of formerly *E. regnans* forest did not naturally regenerate, and while some were replanted at the time [81], limited seed stock meant that some areas were not resown. Much of this area was reburnt in 2009, creating a mosaic of fire age classes and types utilized in this study. This mosaic has also been described in detail by Burton et al. [24] and is comprised of seven field sites of strongly contrasting forest structure and vegetation composition.

**Table 1.** Site information of the instrumented plots and open-weather stations. Site data is located below the relevant open weather station (Powelltown Open & Maroondah Open).

| Site Name | Age [i] | Rainfall (mm $y^{-1}$) | Aspect (°) | Elevation (m asl) | 1759 | 1926 | 1939 | 2009 | 2017 | Coordinates |
|---|---|---|---|---|---|---|---|---|---|---|
| | | | | | \multicolumn | \multicolumn | Disturbance History | | | |
| Powelltown Open | | 1495 | 267 | 740 | | | N/A | | | −37.8992, 145.7310 |
| Acacia$_{10}$ | 10 | 1322 | 134 | 558 | | | | x | | −37.9166, 145.7454 |
| Non-eucalypt$_{10}$ | 10 | 1344 | 128 | 606 | | | x | x | | −37.9133, 145.7459 |
| Non-eucalypt$_{80}$ | 80 | 1402 | 166 | 635 | | x | x | | | −37.9068, 145.7419 |
| *E. regnans*$_2$ | 2 | 1481 | 153 | 735 | | | | | x | −37.9005, 145.7323 |
| *E. regnans*$_{10}$ | 10 | 1337 | 142 | 588 | | | | x | | −37.9148, 145.7452 |
| *E. regnans*$_{80}$ | 80 | 1448 | 204 | 672 | | | x | | | −37.9028, 145.7364 |
| Maroondah Open | | 1318 | 263 | 769 | | | N/A | | | −37.5713, 145.6213 |
| *E. regnans*$_{260}$ | 260 | 1297 | 156 | 727 | x | | j | j | | −37.5728, 145.6161 |

[i] refers to postfire age, from 2019 when most data were collected, [j] low-intensity, non-stand-replacing burn in understory.

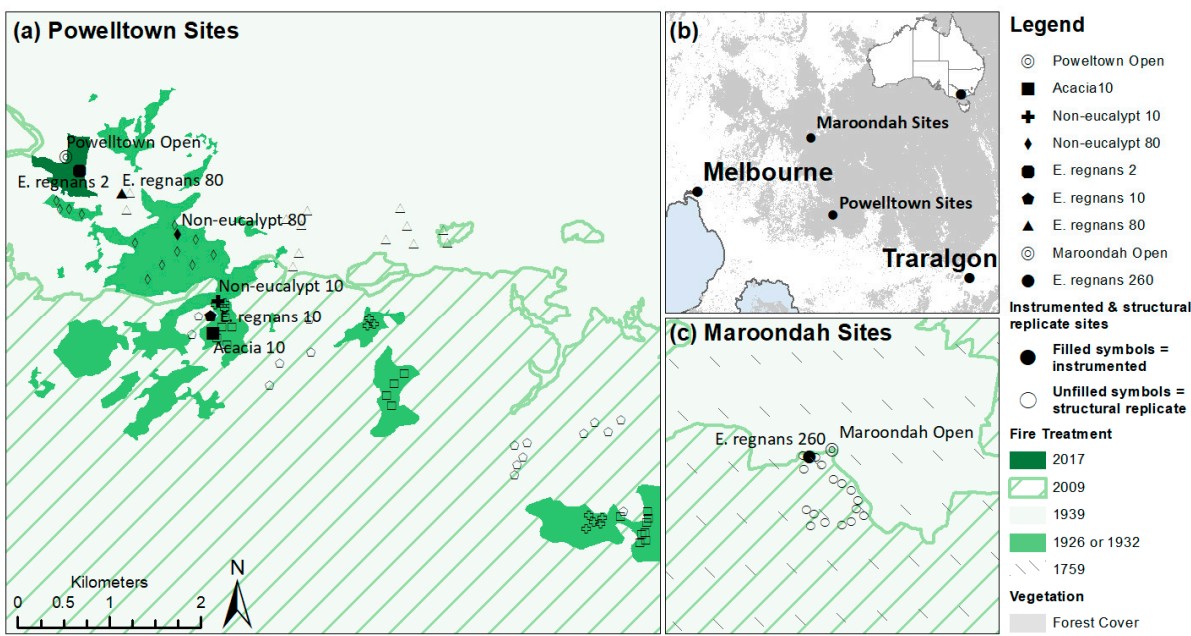

**Figure 2.** Location of the study areas, including seven instrumented field sites and two open (control) weather stations within the Central Highlands, Victoria, in southeastern Australia (**b**). Coloured and dashed areas indicate the fire disturbance con-ditions, denoting the most recent fires affecting forest structure at the alternative state forest sites. The 1939 fire (light green) affected the entire Powelltown (**a**) and Maroondah (**c**) regions.

### 2.2.2. Forest Structure Analysis

Airborne lidar data were commissioned by the state government environment depart-ment and were acquired in 2016 from a fixed wing aircraft, using a Trimble AX60 sensor. Average point density was 4.38 pts m$^{-2}$ returning a full-waveform product, with 0.22 m footprint per return [82]. Forest structural values (Table 2) were derived from one 20 m radius plot at each instrumented forested site, using the lidR and leafR packages in R and its dependents [83]. Plots of forest structure from each instrumented site are depicted in Figure 3.

**Table 2.** Site forest structure and canopy height metrics derived from lidar.

| Site Name | LPI$_{0.5}$ | LPI$_2$ | LAI$_2$ | LPI$_{CANOPY}$ | LPI$_{DELTA}$ | CH$_{95}$ |
|---|---|---|---|---|---|---|
| Acacia$_{10}$ | 0.06 | 0.07 | 2.91 | 0.60 | 0.01 | 17.0 |
| Non-eucalypt$_{10}$ | 0.06 | 0.09 | 2.88 | 0.70 | 0.03 | 13.8 |
| Non-eucalypt$_{80}$ | 0.07 | 0.16 | 2.6 | 0.40 | 0.09 | 19.3 |
| *E. regnans*$_2$ | 0.36 | 0.65 | 0.84 | 0.65 [i] | 0.29 | 4.9 [j] |
| *E. regnans*$_{10}$ | 0.12 | 0.20 | 2.07 | 0.62 | 0.08 | 14.3 |
| *E. regnans*$_{80}$ | 0.07 | 0.16 | 1.46 | 0.6 | 0.09 | 67.4 |
| *E. regnans*$_{260}$ | 0.13 | 0.18 | 1.50 | 0.83 | 0.05 | 52.7 |

[i] 2 m is the canopy base height of E. regnans$_2$, so values match, [j] manually measured in the field.

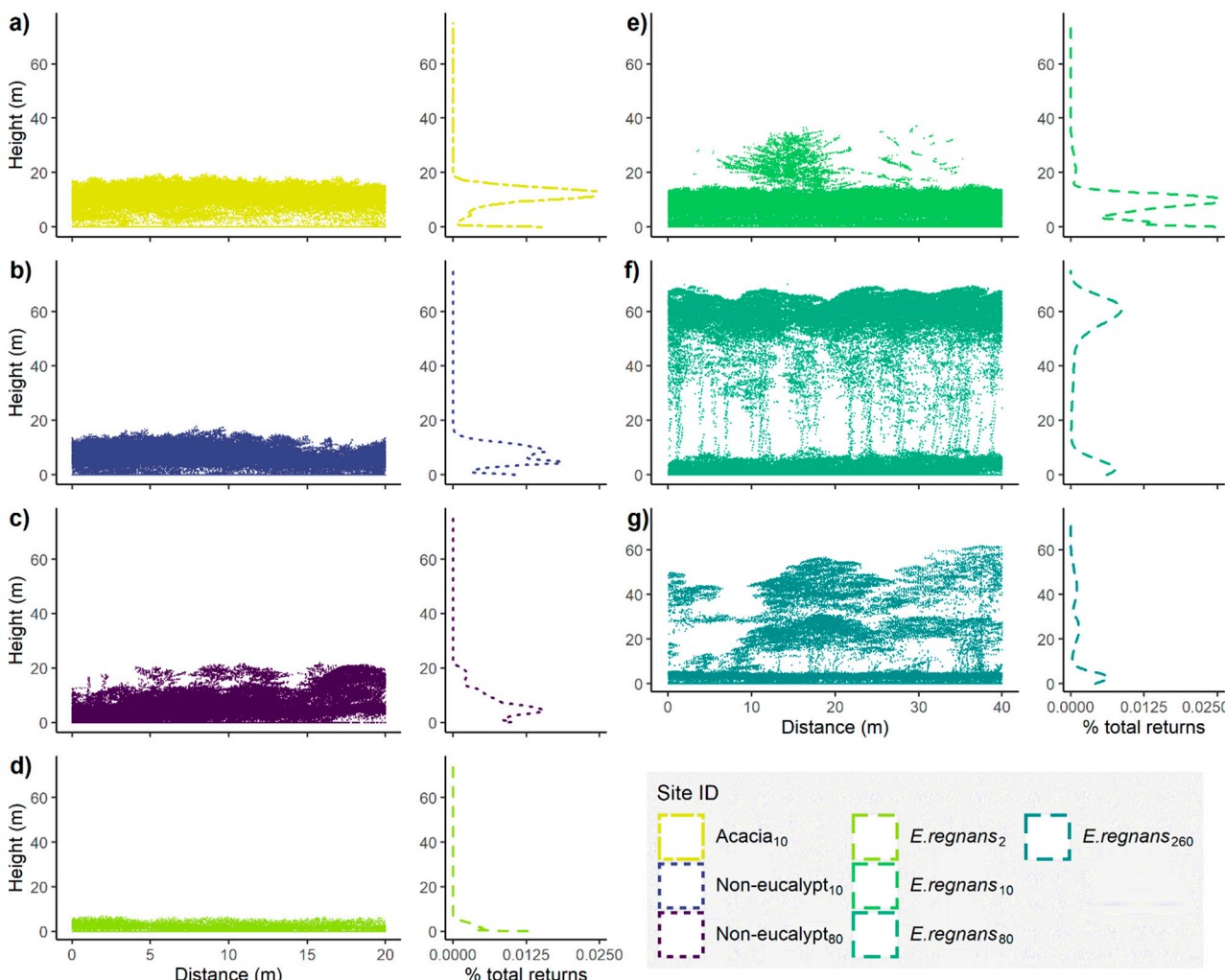

**Figure 3.** Point cloud profiles (left) and percentage return profiles (right) for instrumented field sites: (**a**) Acacia$_{10}$, (**b**) Non-eucalypt$_{10}$, (**c**) Non-eucalypt$_{80}$, (**d**) *E. regnans*$_2$, (**e**) *E. regnans*$_{10}$, (**f**) *E. regnans*$_{80}$ and (**g**) *E. regnans*$_{260}$. The dead stag has not been filtered from *E. regnans*$_{10}$ instrumented site for this figure.

Six metrics describing forest structure were derived from the lidar data (Table 2). The 95th percentile of return height for lidar returns classified as nonground points was taken as the mean canopy height at each site (CH$_{95}$). It was derived using the *quantile* function in R and is a common metric for forest height that moderates the impact of outliers [84]. The *E. regnans*$_{10}$ site contained a mature dead tree (Figure 3d), which was removed from the lidar data by visual inspection and filtering prior to calculating CH$_{95}$. However, because the dead tree provided shade, it was included in leaf area index (LAI) and LPI calculations. LAI was calculated using the *lai* function from the leafR package following Almeida et al. [85]. LPI is the ratio of lidar points that reach a defined height, over the total points in an area [86] and is a proxy for the penetration of solar radiation through the canopy. LPI at 0.5 m (LPI$_{0.5}$) and 2 m (LPI$_2$) were calculated following Bode et al. [52]. LPI$_{DELTA}$ is the difference between LPI$_{0.5}$ and LPI$_2$ and is assumed to be an analogue for the amount of vegetation present between these two heights in the understorey, with a higher value indicating more biomass in this stratum. Canopy LPI (LPI$_{CANOPY}$) was calculated by determining the canopy base height of each instrumented site (through visual inspection of canopy return profiles), then extracting the corresponding LPI value for this base height. It is a proxy for the density of the canopy at each site. The lidar data were collected approximately three years prior to this study, and while small changes to structure are likely over this timeframe, due to the large range of forest structural conditions (Figure 3),

we assume that any changes in structure have a negligible effect on microclimate and FMC comparisions. We did not have lidar available for the *E. regnans*$_2$ site as it was harvested after lidar capture; therefore, forest structure metrics were derived from representative forested sites across the landscape of the same disturbance history. The method used to determine forest structural metrics for this site is detailed in Appendix A.

### 2.2.3. Representativeness of Forest Structure at the Instrumented Sites

Instrumented sites were selected based on differing forest structures due to historical fire disturbance, and consequently, the random allocation of sites was not possible. Replicating the study design was further limited by the cost of the sensor array at each site. Forest structure is the primary difference between sites, and to test if the instrumented sites were representative of other locations with similar disturbance history, we compared forest structural properties estimated from lidar at the instrumented sites to stratified randomly sampled points across the landscape (Figure 2—unfilled symbols). Candidate forest areas for structural comparison were located through field reconnaissance and visual assessment of aerial imagery, and stratified random points allocated in the GIS environment [87]. For the instrumented sites to be considered representative of that alternative states across the broader area, we determined that $CH_{95}$ and $LPI_{0.5}$ should plot within one standard deviation of the 1:1 line. Mean $LPI_{0.5}$ and $CH_{95}$ were derived for each structural replicate site ($n > 12$ per instrumented site, 20 m radius plot) and compared to mean values derived from each instrumented site (Figure 4). Error bars represent one standard deviation (SD) from the mean of the replicate sites. All mean replicate values except one (*E. regnan*$_{260}$, $CH_{95}$) were within one SD of the 1:1 line, demonstrating that forest structure at the instrumented sites can be considered representative of that alternative state forest across the broader landscape. While $CH_{95}$ at *E. regnans*$_{260}$ was less than the landscape mean, given the height of this forest (>60 m) and other studies reporting a negligible effect of canopy height above a lower height threshold [47], we assume that this difference in forest height also has a negligible effect. We did not have lidar available for the *E. regnans*$_2$ site; therefore, mean structural properties were derived from *E. regnans* forests in the landscape regenerated two years prior to lidar capture (Appendix A). At *E. regnans*$_2$, $CH_{95}$ was measured in the field using a structure pole and is compared here (Figure 4a) against the mean landscape value. It was not possible to compare $LPI_{0.5}$.

### 2.3. Field Instrumentation and Data Collection

The control open weather stations recorded air temperature and relative humidity (RH) (CS215 probe, enclosed in a 6-plate 41303-5A shield at 1.5 m, Campbell Scientific Inc., Logan, UT, USA), solar radiation (CS300-L pyranometer, Campbell Scientific Inc., Logan, UT, USA), rainfall (TB-3 tipping bucket gauge, Hydrological Services Pty Ltd., Warwick Farm, NSW, Australia) and wind speed (014A Met One 3-cup anemometer installed at 2 m, Campbell Scientific Inc., Logan, UT, USA). Fuel moisture was recorded using three automated 10 hr moisture sticks (CS506 fuel moisture sensor attached to 26601 10-h stick at 0.3 m, Campbell Scientific Inc., Logan, UT, USA), arranged within 5 m of the centre of each open-weather station. The automated sticks are made of 12.7 mm diameter USFS-standard ponderosa pine dowel and emulate the moisture content of fuels on the forest floor. The forest sites were also instrumented with three automated 10 hr fuel moisture sensors arranged randomly across each site. The sensors were connected to a CR1000 data logger (Campbell Scientific Inc., Log an, UT, USA), which was downloaded to a laptop using LoggerNet software.

We utilised five instrumented sites established by Burton et al. [24], with two additional instrumented sites, and one control weather station subsequently added. The results (Figure 5) are consequently presented in two cohorts, a full timeseries ($n = 420$ days, including 3 summer periods) of the five original sites (Figure 5b,d) and timeseries data that includes all overlapping data ($n = 208$ days) for the additional two instrumented and control weather stations (Figure 5d,e). The time periods of data collection are outlined in

Figure 5a. Reporting in two distinct timeseries maximises the use of available data, while appropriately presenting it for meaningful intercomparison. The modelling approach uses a subset of sequential timeseries data from the seven instrumented sites (*n* = 96). The timeseries data were visually inspected for errors and data spikes associated with sensor errors; there were no missing data or sensor errors in the data presented.

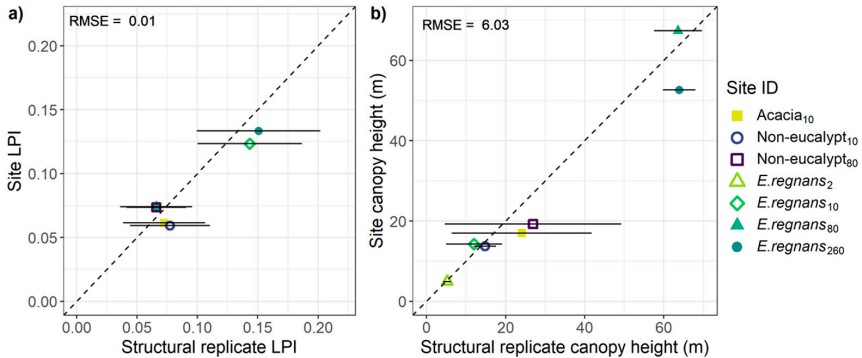

**Figure 4.** Structural replicate plots for (**a**) LPI$_{0.5m}$ and (**b**) 95th percentile canopy height (CH$_{95}$). Instrumented site properties are represented on the y-axis, and structural replicate (*n* > 12) properties on the x-axis. Error bars indicate one standard deviation from the mean of the structural replicate value.

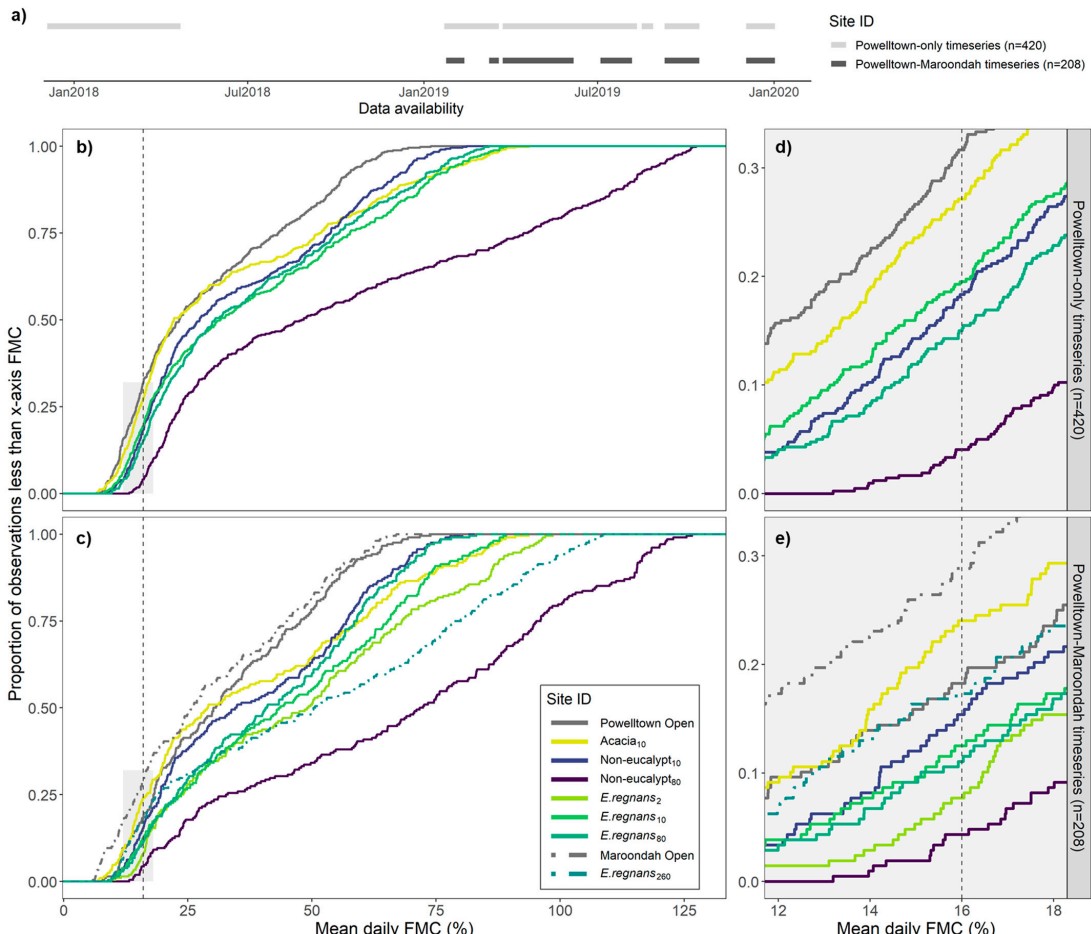

**Figure 5.** Cumulative proportion plots for mean daily FMC, including: (**a**) the timeline of data availability for each timeseries, (**b**) cumulative proportion plot for Powelltown only timeseries across the range of FMC values recorded, and (**c**) Powelltown–Maroondah timeseries, (**d**) critical FMC range for Powelltown only timeseries, and (**e**) critical FMC range for Powelltown–Maroondah timeseries. The dotted vertical line indicates the 16% fuel availability threshold.

*2.4. Data Analysis*

Mean daily FMC (%) was calculated by averaging hourly observations from the three sensors across a 24 h period. We refer to 'fuel availability' in the results, which is defined here as days where mean FMC is less than 16%, which is a threshold value at which fuels are assumed to be dry enough to sustain fire [88]. Daily air temperature and relative humidity values were summarised by calculating the daily mean, maximum and minimum values from hourly sensor readings. Shortwave radiation is presented as daily cumulative radiation load, and precipitation is sum daily precipitation each day. Wind speeds reported are the average and maximum speeds in a 24-h period, derived from hourly sensor readings. Longwave, net radiation and potential evapotranspiration were calculated from microclimate data following Nyman et al. [89] and are presented as daily cumulative loads. All data analyses were conducted in R statistical programming language, version 3.6.3 [90], a summary of data calculated is presented in Table 3.

**Table 3.** Candidate open-weather input variables used in generating the GAMMs.

| Candidate Variable | Value | Denoted as | Unit |
|---|---|---|---|
| Short-wave radiation | Daily sum | SWR | $MJ\ m^2\ day^{-1}$ |
| Air temperature | Daily minimum | $T_{min}$ | °C |
| | Daily maximum | $T_{max}$ | °C |
| | Daily mean | $T_{mean}$ | °C |
| Relative humidity | Daily minimum | $RH_{min}$ | % |
| | Daily maximum | $RH_{max}$ | % |
| | Daily mean | $RH_{mean}$ | % |
| Wind speed | Max | $W_{max}$ | $m\ s^{-1}$ |
| | Mean | $W_{mean}$ | $m\ s^{-1}$ |
| Longwave radiation | Incoming | $LW_{in}$ | $MJ\ m^2\ day^{-1}$ |
| | Outgoing | $LW_{out}$ | $MJ\ m^2\ day^{-1}$ |
| | Net | $LW_{net}$ | $MJ\ m^2\ day^{-1}$ |
| Net radiation | Daily sum | $R_{net}$ | $MJ\ m^2\ day^{-1}$ |
| Precipitation | Daily sum | P | $mm\ day^{-1}$ |
| Vapor pressure deficit | Daily mean | VPD | kPa |
| Potential evapotranspiration | Daily mean | $E_p$ | $mm\ day^{-1}$ |
| Precipitation$_{t-1}$ | Daily sum | $P_{t-1}$ | $mm\ day^{-1}$ |
| $FMC_{site(t-1)}$ | Daily mean | $FMC_{site(t-1)}$ | % |

To determine whether FMC was different across the sites, we generated cumulative density function (CDF) figures for mean daily FMC at each site and timeseries (Figure 5, Table 4). To determine whether FMC could be accurately modelled from open weather, and to test the importance of forest structural properties when estimating mean FMC from open-weather data, we constructed general additive mixed models (GAMMs) using *mgcv* in R [91]. Mixed models allow for the analysis of nested data (whereby FMC is nested by site), while also appropriately treating the fixed effects (open weather to FMC relationship) and random effects (specific effect of site ID on FMC) terms. GAMMS are a nonparametric extension of generalised linear mixed models [92] and are useful when the shape of the relationships between variables and across random effects are not known. Mean daily FMC at each forested site was the dependent variable and all open control weather variables were considered as candidate-independent predictors (Table 3). We included site ID as a random effect to account for site-specific variability. We fitted penalised cubic splines for each candidate variable using restricted maximum likelihood (REML), with the maximum degrees of freedom limited for four to avoid overfitting. Effective degrees of freedom (edf) are reported in Table 5, which measure the complexity of a penalised smooth term. Higher numbers indicate a more dynamic fit [92]. Feature selection was conducted using a stepwise approach, with all candidate variables considered initially and removed if

they were correlated with other predictor variables ($r^2 > 0.80$) or nonsignificant ($p > 0.05$) (e.g., [93]). Candidate variables are outlined in Table 3.

**Table 4.** Summary of the proportion of mean daily FMC values less than 16% relative to the number of days in each timeseries, the proportion less than 16% relative to the local control weather station and the total number of days fuels were available for each timeseries.

| Site ID | Powelltown-Only ($n$ = 420) | | | Powelltown–Maroondah ($n$ = 208) | | |
|---|---|---|---|---|---|---|
| | Proportion Days below 16% | Relative Proportion | Sum Days Available (FMC < 16%) | Proportion Days below 16% | Relative Proportion | Sum Days Available (FMC < 16%) |
| Powelltown Open | 0.32 | 1.00 | 133 | 0.18 | 1.00 | 38 |
| Acacia$_{10}$ | 0.27 | 0.86 | 114 | 0.24 | 1.32 | 50 |
| Non-eucalypt$_{10}$ | 0.18 | 0.58 | 77 | 0.15 | 0.84 | 32 |
| Non-eucalypt$_{80}$ | 0.04 | 0.13 | 17 | 0.04 | 0.24 | 9 |
| *E. regnans*$_2$ | - | - | - | 0.08 | 0.42 | 16 |
| *E. regnans*$_{10}$ | 0.20 | 0.62 | 82 | 0.12 | 0.68 | 26 |
| *E. regnans*$_{80}$ | 0.15 | 0.47 | 63 | 0.11 | 0.61 | 23 |
| Maroondah Open | - | - | - | 0.29 | 1.00 | 60 |
| *E. regnans*$_{260}$ | - | - | - | 0.17 | 0.60 | 36 |

**Table 5.** Diagnostics of the GAMMs developed with (a) independent (weather-only) data and (b) autoregressive data which included lagged variables at the site level.

| | Smoothing Function | edf | Fisher Test | *p*-Value |
|---|---|---|---|---|
| **(a)** | Relative humidity (%) | 2.505 | 10.536 | *** |
| | Max wind speed (m s$^{-1}$) | 2.339 | 15.535 | *** |
| | Outgoing longwave radiation (MJ m$^2$ day$^{-1}$) | 2.847 | 138.036 | *** |
| | Net radiation (MJ m$^2$ day$^{-1}$) | 1.702 | 2.854 | *** |
| | | | | |
| **(b)** | Relative humidity (%) | 2.082 | 21.321 | *** |
| | Max wind speed (m s$^{-1}$) | 2.805 | 9.504 | *** |
| | Outgoing longwave radiation (MJ m$^2$ day$^{-1}$) | 2.729 | 31.674 | *** |
| | Net radiation (MJ m$^2$ day$^{-1}$) | 1.739 | 4.178 | *** |
| | Previous day open precipitation (mm day$^{-1}$) $_{(t-1)}$ | 2.546 | 38.741 | *** |
| | Previous day subcanopy FMC (%) $_{(t-1)}$ | 2.907 | 756.999 | *** |

Significance codes: $p < 0.001$ = ***.

Following Bradshaw et al. [67] we expect the automated fuel sticks to reach equilibrium within a 24-h period. However, the response time of fuel sticks in forested settings has been poorly evaluated. To account for this, we constructed two GAMMs to ensure any effects of lag in fuel moisture response in the system were captured. One model utilised open weather on the same day only (GAMM$_{INDP}$), while the other included autoregressive weather and subcanopy terms (e.g., [94]) from the previous day (GAMM$_{AR}$). The GAMM random effects intercepts represent the degree to which the model is shifted up or down the y-axis to predict site-specific FMC values from the control weather inputs. Because forest structural properties are the only meaningful differences between the alternative state and open sites at the daily level, in practice, this intercept represents the degree to which site forest structure moderates mean daily FMC. Linear regression was used to evaluate the relationship between random effects model intercepts and instrumented site forest structural metrics and to infer relative importance of these structural elements.

## 3. Results

### 3.1. Differences in Fuel Availability

In both timeseries, there were distinct concordant differences in fuel availability between the end-member sites; Acacia$_{10}$ and noneucalpyt$_{80}$, which had the highest and lowest fuel availability respectively (Figure 4). However, the fuel availability ranking of sites in the mid-range varied across the two timeseries presented. In the Powelltown-only timeseries, the relative proportion of fuel availability (RFA) at the instrumented sites compared to the control weather station (Table 4) ranged from 0.13 (Non-eucalypt$_{80}$) to 0.86 (Acacia$_{10}$), equating to a difference of 97 days across the timeseries. The fuel availability rankings for end-member sites were mirrored in the Powelltown–Maroondah timeseries (Acacia$_{10}$, RFA = 1.32; Non-eucalypt$_{80}$, RFA = 0.24), although the difference in relative fuel availability increased between the two timeseries presented. At the 16% fuel availability threshold, fuels at Acacia$_{10}$ were more available than the Powelltown control site (Figure 5e).

### 3.2. Modelling Understorey FMC from Open Weather

To understand the subcanopy processes driving FMC in wet forest systems, we constructed two GAMMs using candidate variables in Table 3. The best-performing independent model (r$^2$ = 0.551) had four open-weather predictors ($p < 0.001$), although the addition of two autoregressive terms substantially improved the skill of the model (GAMM$_{AR}$ r$^2$ = 0.936). GAMM$_{AR}$ included the four predictors in GAMM$_{INDP}$ with two additional autoregressive terms: FMC$_{site(t-1)}$ and P$_{t-1}$, all significant at the $p < 0.001$ level (Table 5). Site was included as a random effect in both models and was significant at $p < 0.001$ level, demonstrating the importance of site properties in predicting FMC across alternative forest states. Partial plots of the penalised regression splines were developed for both models and illustrate the effect of each predictor on FMC. The partial plots for GAMM$_{INDP}$ are presented in Figure 6 and the partial plots for GAMM$_{AR}$ in Figure 7.

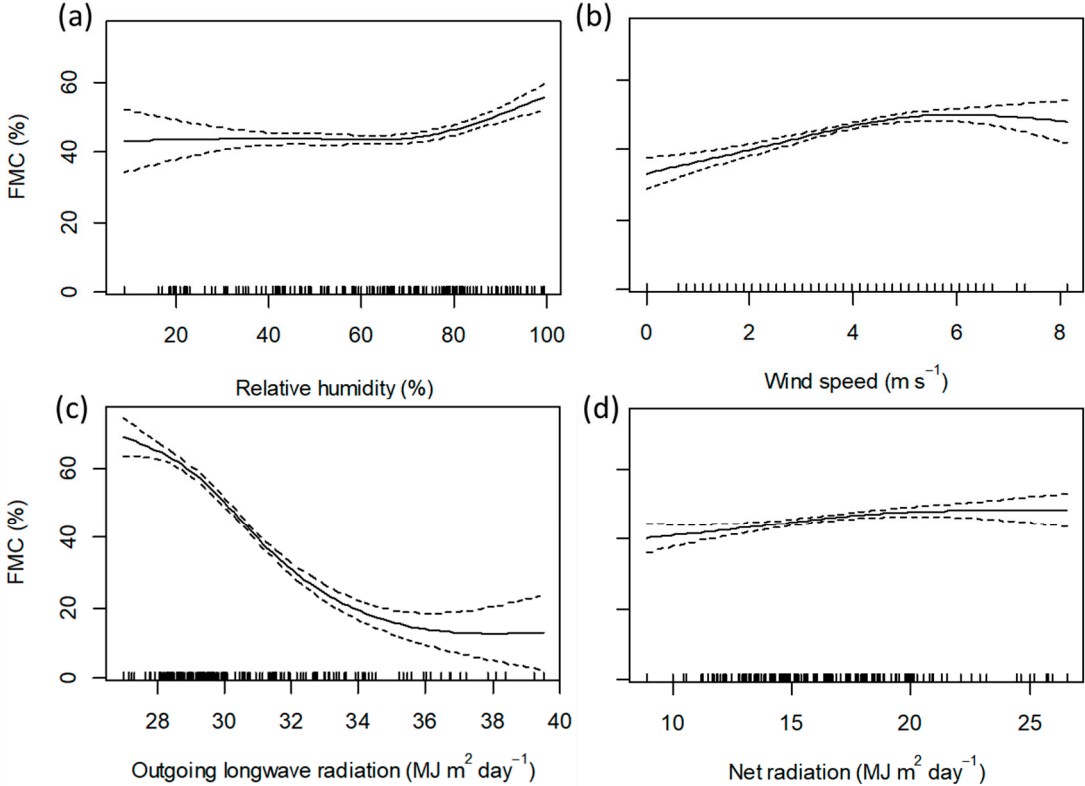

**Figure 6.** Partial plots of GAMM$_{INDP}$ fits for mean daily FMC (%): (**a**) relative humidity, (**b**) maximum wind speed, (**c**) outgoing longwave radiation and (**d**) net radiation. The plots are centered on mean FMC with 95% confidence intervals.

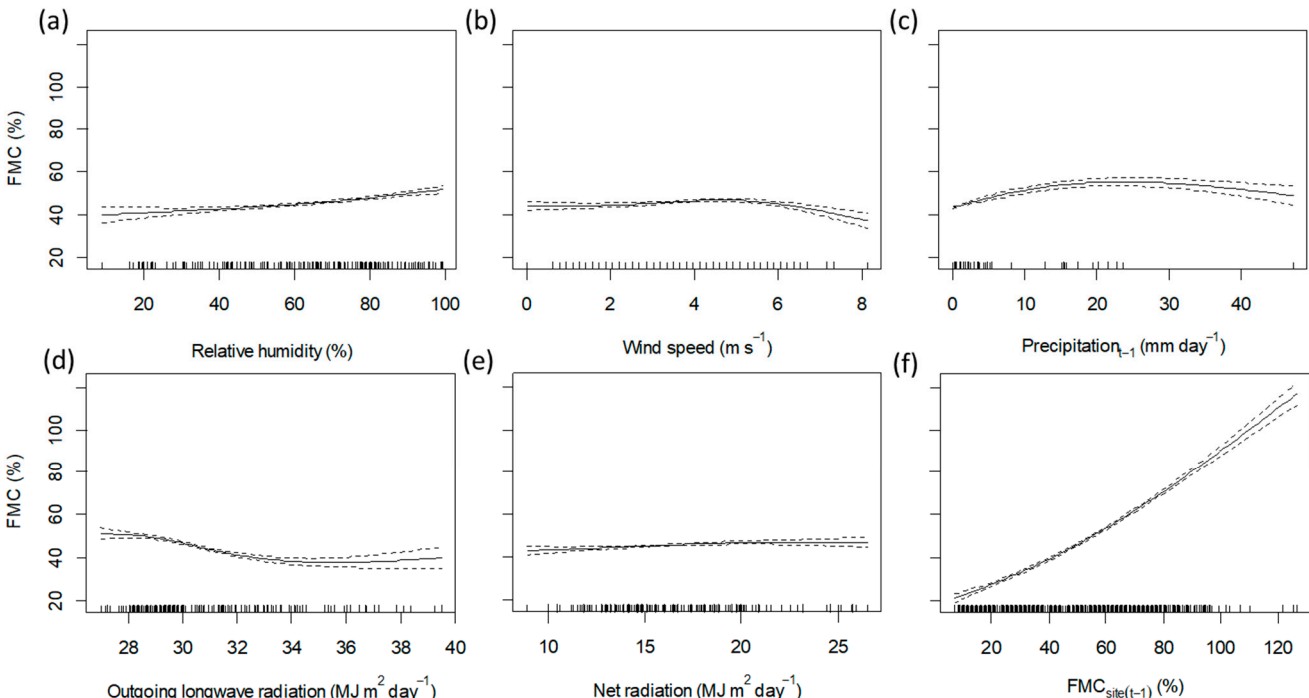

**Figure 7.** Partial plots of $GAMM_{AR}$ fits for mean daily FMC, (**a**) relative humidity, (**b**) maximum wind speed, (**c**) precipitation$_{t-1}$, (**d**) outgoing longwave radiation, (**e**) net radiation and (**f**) $FMC_{site(t-1)}$. Plots are centered on mean FMC with 95% confidence intervals.

### 3.3. Influence of Forest Properties on Understorey FMC

There were distinct differences in forest structural properties (Table 2) and profiles (Figure 3) across the seven instrumented sites. $CH_{95}$ ranged from 4.9 m at *E. regnans*$_2$ to 67.4 m in *E. regnans*$_{80}$, the shortest and tallest sites, respectively. The lowest $LPI_{0.5}$ (0.06) and $LPI_2$ (0.07) values were recorded at $Acacia_{10}$, which also showed the smallest change in LPI across these two heights ($LPI_{DELTA}$ = 0.01). $LPI_{0.5}$ and $LPI_2$ were both highest at *E. regnans*$_2$, recording 0.36 and 0.65, respectively, despite having the second-lowest fuel availability ranking. $LAI_2$ ranged from 0.8 at *E. regnans*$_2$ to 2.9 at $Acacia_{10}$.

$GAMM_{INDP}$ site random effects intercepts were plotted against forest properties to evaluate their relationship with mean daily FMC (Table 6). The strongest correlations for $GAMM_{INDP}$ were with $LPI_{CANOPY}$ ($r^2$ = 0.45) and $LPI_{DELTA}$ ($r^2$ = 0.43), while the weakest correlations were with $LAI_2$ and time since disturbance (Figure 8). The autocorrelated terms were the dominant predictors in $GAMM_{AR}$, making meaningful interpretation of correlations between the random effects intercepts and forest structure difficult, because variance cannot be appropriately apportioned between the predictors. While it cannot be appropriately correlated with a forest structural metrics (and so is not discussed further in this context), the significance of the lag term in improving the performance of the GAMM is a key finding in this research.

**Table 6.** Correlation coefficients of linear regressions between GAMM$_{INDP}$ random effects intercepts and forest structural properties.

| Forest Property | $r^2$ |
| --- | --- |
| Time since disturbance | 0.07 |
| CH$_{95}$ | 0.22 |
| LAI$_{2m}$ | 0.04 |
| LPI$_{0.5m}$ | 0.22 |
| LPI$_{2m}$ | 0.32 |
| LPI$_{DELTA}$ | 0.43 |
| LPI$_{CANOPY}$ | 0.45 |

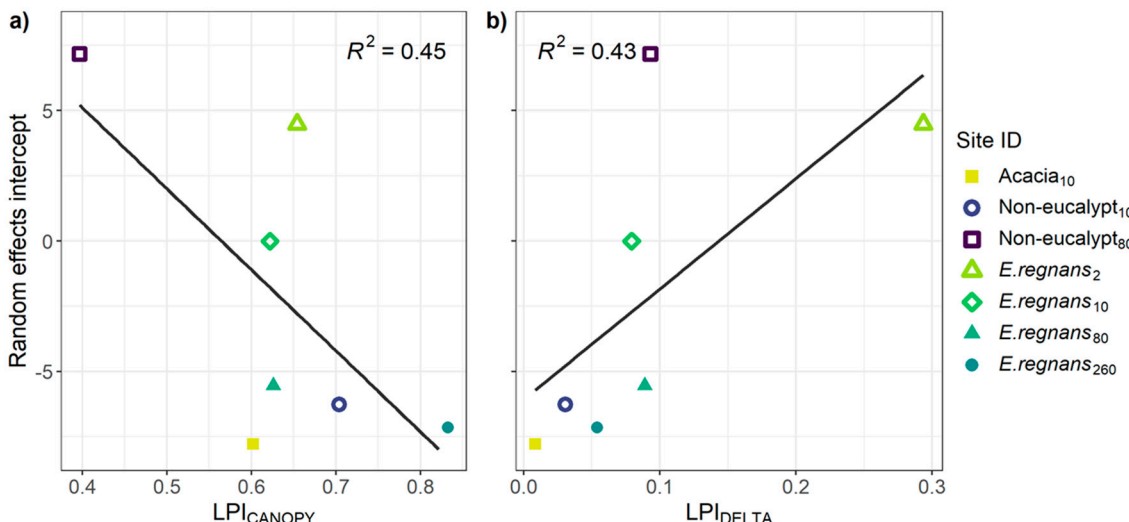

**Figure 8.** Simple linear regressions plots of the best performing forest structural properties to predict GAMM$_{INDP}$ random (site) effects intercepts for (**a**) LPI$_{CANOPY}$ and (**b**) LPI$_{DELTA}$.

## 4. Discussion

### 4.1. Differences in Fuel Availability across Alternative Forest States

The results show that forest structure exerts a strong control on dead fuel moisture across alternative forest states. Distinct differences in days of fuel availability demonstrate that FMC variability translates into meaningful differences in the potential for fire ignition and spread. Our results aligned with previous studies across these sites [24] in regard to moisture availability at the extreme end-members (highest and lowest). However, site rankings of fuel availability in the mid-range varied between the two timeseries, suggesting that fuel availability at the site level is a function of daily weather, in additional to antecedent climate conditions that were not captured by our instantaneous or 24 h lag statistical modelling approach. Fuels were most available at Acacia$_{10}$, which is an alternative vegetation state after recurrent high-intensity fire in mature *E. regnans* stands (Figure 1), and this finding supports the existence of short-term, positive fire–vegetation feedbacks. Fuels were least available at Non-eucalypt$_{80}$. This finding, in conjunction with there being few examples of Non-eucalypt$_{80}$ forest across the landscape, suggests that this alternative forest state is likely not stable and is potentially an example of a negative or stabilising fire–vegetation–climate feedback.

Many empirical- [24], conceptual- [95] and remote sensing-based [96] studies of fuel moisture and availability describe strong age-related differences in fuel availability, with younger forests suggested to be available to burn more often than mature or old growth forests in the *E. regnans* niche. In our study, while the three 10 year-old sites were available more frequently than the other alternative states, there was considerable variability across the two timeseries demonstrating uncertainty related to single-season studies of fuel

moisture in these forests. We also observed divergence in FMC response across the two dominant succession types. In this climate-affected system, we expect that over time, Acacia$_{10}$ and Non-eucalypt$_{10}$ would transition to Non-eucalypt$_{80}$ (Figure 1). We found strong age-mediated fuel availability relationships within these alternative states, where fuel availability was consistently higher in 10 year-old forests compared to mature non-eucalypt forest (Table 4). However, the results were less distinct in *E. regnans* forests, where fuel availability did not change linearly with age. Fuel availability was low at the *E. regnans*$_2$ site, and while relative availability was highest at the *E. regnans*$_{10}$ site, fuel availability plateaued after this point rather than declining as some studies have suggested for mature and old growth forest [96]. Cawson et al. [97] reported that stand age is not a key driver in FMC for forests older than 33 years, and given limited difference in fuel availability between forests regenerated in 2009 (*E. regnans*$_{10}$, RFA = 0.68) and 1759 (*E. regnans*$_{260}$, RFA = 0.60), our research suggests that dead fuel moisture insensitivity to time since disturbance may begin earlier than 33 years.

A limitation of our study is that we only investigated dead FMC using automated fuel moisture sticks. Across the range of alternative forest states, live FMC, which is almost always much higher than dead FMC [98], may play an important role in site-scale fuel moisture dynamics. In addition, different physical and chemical properties of the litter across alternative state sites may alter fuel availability [99]. While standardised automated fuel moisture sticks provide a useful metric for intercomparison across states, drying processes for the ponderosa stick and leaf litter properties unique to each alternative state may vary and therefore be a source of error in our analysis.

### 4.2. Modelling Understorey FMC from Open Weather

Our modelling approach assumed no prior knowledge of macro- or micro-climate effects on FMC in wet forests, which was appropriate given the uncertainty regarding the performance of existing FMC models in wet forests. RH$_{mean}$ and W$_{max}$ were significant predictor variables of FMC in both GAMMs (Figure 6), which are commonly included in models of dead FMC [100]. RH represents the amount of water vapour in the air as a percentage of the total amount at saturation [101] and is buffered by forest canopies to be generally higher in the understorey than in the open [42], and in wet forests, between near-surface and screen height (1.5 m) [102]. Wind speed is typically lower under tall dense forests [103], which limits the lateral transfer of moisture through advection. Nyman et al. [89] reported that high aerodynamic resistance in the subcanopy of wet eucalypt forests limited moisture loss through advection. However, a key difference in our study is the broader range of forest structural conditions investigated. Forest profiles uniquely partition wind in the understorey [54], and the broader range of forest types presented in this study may have generated a larger gradient of subcanopy wind conditions, leading to the significance of wind speed in the model compared to Nyman et al. [89]. The effect of wind speed on FMC was not a negative relationship as we might expect; rather, as wind speed increased so too did FMC. We expect that this is due to higher mean wind speeds in the spring and autumn periods, which are also associated with seasonally wetter fuels. The two other significant variables common to GAMM$_{INDP}$ and GAMM$_{AR}$ were R$_{net}$ and LW$_{out}$, which are the primary drivers of surface evaporation in wet forest systems [89,104]. LW$_{out}$ was the strongest microclimate predictor, consistent with other research in wet forests [89,105,106]. The dense canopy in wet forest attenuates incoming shortwave radiation [107], which can increase the magnitude and relative importance of longwave energy components [108]. However, these components of the energy budget are rarely included in FMC models, and we contend that including these variables constitutes a significant advance in modelling FMC in wet forests.

A key finding of this research is that in wet forests the FMC model which included autoregressive terms for precipitation and fuel moisture greatly outperformed that using daily data alone, which was also noted by Lee et al. [105] for precipitation only. Of these autoregressive variables, FMC$_{t-1}$ had the strongest influence on mean FMC (Figure 7).

Numerous FMC models exist for specific forest types, conditions, or temporal and spatial scales with approaches typically based on vapour exchange principles [32,109–112], semi-physical account-keeping approaches (e.g., [70,113–115]) or fully process-based [71]. Despite this substantial knowledge base, there is no established paradigm for the conditions under which an account-keeping or instantaneous model is most appropriate. In drier forests, with limited effects of canopy buffering, instantaneous models may provide robust estimates of FMC [73]. However, complex forest structure in wet forests decouples the understorey fuels from the prevailing weather conditions [43] and from other understorey strata [102]. At the same time, dense canopy cover and soils with high water-holding capacity, which are typical of wet forests, potentially facilitate temporal buffering of conditions through site soil moisture-humidity feedbacks [42,49,116].

Ten hour automated fuel sticks are widely used operationally by fire managers [66], and while the response time has been extensively studied under standard and open conditions, it is poorly evaluated in forested environments. Substantial improvement between the independent and autoregressive models in this research demonstrates that the lag for the fuel stick under dense canopies is larger than expected (Table 5). In turn, this suggests that there is a source of lag in the system separate to that accounted for through the reported 10 h lag, which may be introduced through forest structure-mediated temporal buffering of the microclimate, which has been reported to occur in forests globally [46]. We posit that temporal buffering capacity of dense forests in this study ultimately led to nonindependence of FMC observations at the daily level, evidenced by the strong lag effect ($FMC_{site(t-1)}$) in the $GAMM_{AR}$ model. Consequently, account-keeping approaches to FMC estimation are likely to be most appropriate in wet forests.

### 4.3. Influence of Forest Structural Properties on FMC across Alternative States

Site random effects (a proxy here for forest structure) were significant when modelling understorey FMC from open weather. This result demonstrates that quantifying the effect of forest properties on macro- to micro-climate is critically important for accurate representation of FMC when modelling across forested landscapes from open-weather data, which is not currently explicitly accounted for in FMC models used operationally in southeastern Australia [32]. The strongest correlations between the modelled site intercept and forest structure were for $LPI_{CANOPY}$ and $LPI_{DELTA}$, which explained 45% and 43% of the variability in FMC, respectively (Figure 8; Table 6). LPI in the canopy varied from 0.4 to 0.83 and, contrary to our expectations, was not linearly or positively associated with fuel availability. We calculated $LPI_{DELTA}$ to evaluate whether changes in light transmittance (also analogous to the amount of vegetation) close to the height where FMC is recorded contributed to fuel availability in these systems, as others have shown strong microclimatic gradients across this domain [102]. The largest change in LPI between 0.5 and 2 m was observed at *E. regnans*$_2$ (Table 2, $LPI_{DELTA}$ = 0.29), which had the second-lowest fuel availability. However, 2 m was also the height of the base of the canopy at the juvenile site ($CH_{95}$ = 4.9 m); therefore, $LPI_{DELTA}$ represents the change in light penetration between the canopy and 0.5 m, not just understorey strata, which is likely a source of error in our analysis contributing to the large change in LPI in this metric. When the *E. regnans*$_2$ is excluded, Acacia$_{10}$ (0.01) and Non-eucalypt$_{80}$ (0.09), which are the end-member fuel availability sites, also have the smallest and largest amount of vegetation between 0.5 m and 2 m respectively. Higher understorey density at Non-eucalypt$_{80}$ may point to higher rainfall interception, transpiration and surface roughness in this stratum. We suggest that these effects would correspondingly increase humidity at the fuel-boundary-layer interface, which may explain the extreme differences in FMC between Non-eucalypt$_{80}$ and Acacia$_{10}$, which had limited understorey vegetation.

While positive (negative) relationships between canopy cover and fuel moisture (fuel availability) have been reported in temperate eucalypt [97], tropical [117] and conifer forests [118], other studies have reported limited effects of cover-mediated changes to fuel moisture [119–121]. Our results are more closely related to the latter of these studies, where

no strong linear results between fuel availability and canopy cover were observed. This may be related to the comparatively small climatic gradient across our sites. Additionally, the effects of canopy cover on FMC can interact with other factors such as topographic position (e.g., slope and aspect [89]), season [122] and site hydrology [42], and, as our results indicate, may depend on where in the profile forest structure is apportioned. Collectively, these factors may explain divergence across the literature.

*4.4. Implications for Forest Flammability*

Our study affirms previous research findings that alternative forest states have divergent fuel moisture responses after high-severity fire [24], with potential for positive or negative fire–vegetation feedback processes depending on the type of alternative forest and its age. Critically, we have determined that the effect of prevailing weather on fuel availability is strongly associated with not only the density of the canopy (through $LPI_{CANOPY}$) but is also a function of the density of vegetation in the understorey ($LPI_{DELTA}$)—which varies substantially across the alternative forest states investigated in this study. This study was undertaken in forests that currently are, or are alternative states to, *E. regnans* forest, which are highly valued systems in this landscape. The potential for these forests to transition to alternative states and their maintenance (or lack thereof) through positive or negative feedbacks have important implications for biodiversity, socio-cultural and economic forest values.

Our study focused on fuel moisture content, which is an important flammability switch in wet forests [34]. However, the propensity for high-consequence wildfire also depends on other elements of flammability, such as an ignition source [123], fuel load and arrangement [124] and flammability traits of dominant species [99,125]. We only measured one element of flammability, which is a limitation of our study. While we determined that $LPI_{DELTA}$ had an important role in fuel availability across alternative states, it is also a metric analogous to fuel load in this stratum (0.5–2 m). In the context of alternative forest states, interactions between fuel moisture and fuel load may have opposing effects on flammability, with important implications for the strength of fire–vegetation feedbacks in this system. For example, low $LPI_{DELTA}$ (and therefore fuel load) at $Acacia_{10}$ was correlated with high fuel availability, and high $LPI_{DELTA}$ (high fuel load) at $Non\text{-}eucalypt_{80}$ with low fuel availability. This contrast characterises the fuel production/fuel dryness flammability switch described by Bradstock et al. [34], and while we know that this switch exists across a landscape aridity gradient [126], our research provides a case study example of this dichotomy within an aridity class across a forest biomass gradient as a result of fire–vegetation feedbacks. For example, while $Acacia_{10}$ has high fuel availability, low biomass at the surface ($LPI_{DELTA}$) suggests that fuel load may limit flammability in this alternative state forest. Future research should focus on investigating the interactive role of multiple flammability factors on vegetation–feedbacks, not only fuel moisture content, in these complex landscapes.

We successfully modelled subcanopy FMC from open-weather stations, while demonstrating the importance of site-specific forest structure variables such as light penetration index to the performance of the model. Current approaches to estimating FMC in operational applications utilise the moisture content submodel in McArthur's forest fire danger index (FFDI) [32], with input values of temperature and relative humidity. The input data are typically sourced from landscape-scale gridded weather products generated from open-weather stations [61] that do not include adjustments accounting for the effect of forest structure, which we know to be important [127]. In wet forests, temporal buffering creates a lagged FMC effect, such that daily observations of FMC in wet forests are not independent. While simple empirical models of FMC have been extraordinarily successful despite the simple inputs and lack of subcanopy transfer functions, in wet forests, these models are likely to generate erroneous, or highly variable FMC predictions. Our research suggests that fire managers should investigate the landscape scale applicability of process-

based account-keeping FMC models, with weather data scaled by forest properties to the microclimate scale.

## 5. Conclusions

Recurrent high-intensity fire alters community composition and structure in obligate seeder forests, facilitating shifts to alternative forest states. Our results demonstrate that understory dead fuel moisture in wet forests is significantly affected by forest structural properties across these alternative states, including a substantial lag effect, to an extent that is meaningful for fire ignition and spread. Lidar-derived forest structural metrics related to understorey and canopy density were found to be important predictors moderating the relationship between open weather and fuel moisture across alternative states, and differences in these structural metrics led to observations of both positive and negative feedback processes. Our results constitute an important step towards the identification and quantification of fire–vegetation feedbacks in wet forests that may amplify or dampen the direct effects of climate change on fire activity, which has significant implications for the resilience of these vulnerable and valuable obligate seeder forests.

**Author Contributions:** Conceptualization, T.P.B., J.B. and G.J.S.; data curation, T.P.B., J.B., P.J.N.; fieldwork and data collection, T.P.B., J.B. and P.J.N.; formal analysis, T.P.B.; funding acquisition, P.N.J.L. and G.J.S.; methodology, T.P.B., J.B., A.I., T.J.D. and G.J.S.; supervision, T.J.D., A.I., G.J.S. and P.N.J.L.; visualization, T.P.B.; writing—original draft, T.P.B.; writing—review and editing, T.P.B., A.I., T.J.D., G.J.S., P.N.J.L., J.B. and P.J.N.; funding acquisition, G.J.S. and P.N.J.L. All authors have read and agreed to the published version of the manuscript.

**Funding:** This research was funded by an Australian Government Research Training Program (RTP) Scholarship, the Integrated Forest Ecosystem Research (IFER) Agreement between the Department of Environment, Land, Water and Planning and the University of Melbourne (THEMIS TA37690), and Melbourne Water (THEMIS 301102). TPB received a travelling scholarship from the Institute of Foresters Australia (IFA) in 2019.

**Data Availability Statement:** The FMC and weather data presented in this study are available on request from the corresponding author. The lidar data are publicly available from the Victorian Government.

**Acknowledgments:** This research was conducted on the unceded lands of the Wurundjeri and Taungurung people, the authors acknowledge and pay our respects to their Elders past and present. This research was conducted in State forest under DELWP permit number FS/14/3694/1. We thank Joe Hall, Chris Lyell, Emma Keith and Sam Hillman for assistance with site installation and data collection. We gratefully acknowledge and thank three anonymous reviewers, who provided valuable feedback on the manuscript.

**Conflicts of Interest:** The authors declare no conflict of interest.

## Appendix A

*E. regnans$_2$ Forest Structure Data*

The lidar data were captured in 2016, while the *E. regnans$_2$* was harvested in 2017. Consequently, we did not have lidar data at this site. To derive proxy forest structural properties for *E. regnans$_2$*, we stratified the landscape using a logging history spatial layer to identify coupes regenerated in 2014, which was two years prior to the lidar being flown. In total, 16 coupes of a similar aspect and age range were identified across the Central Highlands, and 183 points were randomly sampled from within those coupes to derive proxy forest structural metrics for *E. regnans$_2$*. Resultant data were logic-checked using site measurements, aerial photography [128] and an assumed mean annual growth increment of 2.1 m pear year [129].

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
