# Peer review of "Forest Structure Drives Fuel Moisture Response across Alternative Forest States"

_fire, doi:10.3390/fire4030048_

Round 1

Reviewer 1 Report

I would like to commend the authors for a very nice study on how different alternative vegetation states could affect DFMC, potentially exerting negative and positive feedbacks. I would just suggest the authors to clarify from the start that the manuscript is focused on DFMC. Also, in the GAM plots, please show the actual fit (actual FMC on y-axis)

Reviewer 2 Report

Brown et al. submit a manuscript reporting in wet forests the importance of forest structure in estimating fuel moisture content (specifically dead fuel), considering the alternative forest states associated to fire. The manuscript is well stated and clear, however some details can be improved.

Minor comments:

Line 64: Since through all the manuscript you mention the scientific name for E. regnans, would be better to present common name, mountain ash, between parenthesis.

Lines 159-167: when presenting your objectives and questions I think you should re-state that you will be considering specifically dead fuel, and not live fuel.

Figure 2: would be nice if you include letters to each sub-figure, and if you present first (as “a”) the general map, in the top left corner, below Marrondah sites, and on the right the detail for Poweltown sites. In Poweltown sites detail you mention Acacia 10 as A. dealbata 10, you should unify, in concordance also with text (I think you always use Acacia 10, but please check). 1939 identification color is a little difficult to clearly visualize in the figure, could this be improved?

Line 195: should it be “dashed”?

Table 1: specify in footnotes or in legend that “age” refers to “post-fire age”  

Lines 234-236: is not completely clear if forest structural values were derived from only 1 20m2 plot at each instrumented site and also at each replicated site? Clarify. How many plots per type of habitat did you considered?

Figures and tables in general: it would be easier for the reader if you use for all tables and figures the same order of presentation of sites in the legends (or order in tables).

Figure 3: It seems that letters in the figure for each sub-figure do not correspond with details in legend. Please check

Lines starting in 281: I suggest to include a/some line/lines explaining specifically what do you refer with “instrumented” and “replicates” and the differences between these sites, which measurements did you made at each type of site for example. Information is given along materials and methods, but it is not given clearly and straightforward.

-Along the manuscript you refer to the different types of sites as “treatments”. Technically, these are not treatments per se, they are “conditions” or “types of sites”. You are not manipulating any experimental treatment. I suggest to modify this.

-Lines 326 and 331: check for the error of references links.  

-Line 396: Despite it might be implicit, it would help for the reader to specify that you refer specifically to Poweltown control site

-Line 417: According to your general results and discussion, it would be better to see in the manuscript the partial plots for GAMMar, and plots for GAMMindp in appendix

-Order numbers for table 4, 5 and 6 are not correct in the legends. Please check if they are correct in the manuscript text.

-Line 430: “values were recorded AT Acacia10,…”

-Line 453: change order for “state forests”… should it be “forests states”?

-First paragraph of discussion: Across first lines I think you should re-state again (in some few words) that your results of “fuel availability” are inferred from DEAD fuel moisture content threshold.

-Lines 476-477: there is something missing in this sentence?

-Line 488 and paragraph starting in line 533: Despite I agree it is a great methodology for making global comparisons, you should also mention as a limitation that standardized automated ponderosa sticks, surely may not have the same physical-chemical tissues/material properties than main dominant species at your sites, and hence the process of moisture loss might have slight differences, between ponderosa and your species, and also between dominant species at each site.

-Lines 510 and 582: after reading these considerations in the discussion, about seasonality, I would have expected to see some analyses or exploration between all your results considering specifically the different seasons that were included in your time-series. What about your results (regressions, models) if you analyze separately the season of historically maximum probability of fire occurrence? Would this reinforce your models? Perhaps the inclusion of the periods of non-fire seasons from your time-series might be diluting the effect of forest structural properties across types of sites?   

Reviewer 3 Report

This manuscript provided an interesting topic to investigate the importance of forest structure in estimating FMC. Lidar-derived forest structural metrics related to understorey and canopy density were found to be important predictors moderating the relationship between open weather and fuel moisture across alternative states. However, this work needs substantial revision to address some critical points as listed below.

(1) Page 9, Line 338: “Mean daily FMC(%) was calculated by averaging hourly observations from the three sensors across a 24-hour period.” Daily minimum dead fuel moisture is a key determinant of fire (de Dios et al. 2015), why don’t use this term? Since hourly fuel moisture was recorded by CS506, why don’t use it?

(2) Page 10, Line 369: “r2>0.80”. How do you determine 0.8, why not 0.7 or 0.9?

(3) Page 14, Figure 7: The number of sites is too limited, which may lead to the results be uncertain. For example, the site Non-eucalypt80 is crucial to the linear regression result of LPICANOPY and random (site) effects intercepts. While the site E. regnans2 is crucial to the linear regression of LPIDELTA and random effect intercepts.

(4) Page 15, Line 493-532: GAMMAR has two variables than GAMMINDP, can you make sure the importance of the variables. Which is more important, Pt-1 or FMCsite(t-1)?

(5) Page 16, line 547-554: Considering both canopy and understorey, how do you make sure the random effect intercepts?

(6) Page 16, line 571-574: The best-performing dependent model (r2 = 0.551) had four open weather predictors, which were measured at screen height (RH,1.5m; wind speed, 2m; solar radiation, 2m). All the sensors are above understorey, therefore, the measured variables may have been affected by understorey, which may bring errors to the linear regression of LPIDELTA and random effects intercepts.

(7) The manuscript seems too long to be readable. Is that possible for the author to make this manuscript terser, and put some materials to a Supplementary Material for example? This is only a minor comment and suggestion.
